# Distilling human decision-making dynamics: a comparative analysis of low-dimensional architectures

**Hua-Dong Xiong**[*]
Department of Psychology
University of Arizona
hdx@arizona.edu

**Li Ji-An**[*]
Neurosciences Graduate Program
University of California, San Diego
jian.li.acad@gmail.com

**Marcelo G. Mattar**[†]
Department of Psychology
New York University
marcelo.mattar@nyu.edu

**Robert C. Wilson**[†]
Department of Psychology
University of Arizona
bob@arizona.edu

## Abstract

Recent advances in examining biological decision-making behaviors have increasingly favored recurrent neural networks (RNNs) over traditional cognitive models grounded in normative principles such as reinforcement learning. This shift owes to RNN's superior predictive performance on behavioral data, achieved with minimal manual engineering. To glean insights into biological decision-making through these networks, this approach focuses on identifying a compact set of latent dynamical variables by restricting the size of the recurrent layer's bottleneck. Yet, little is known about the distinctions between these low-dimensional RNN architectures and their practical effectiveness in capturing behavioral patterns of biological decision-making. Our study bridges this knowledge gap by 1) offering an architectural comparison of these low-dimensional RNNs with standardized terminology; 2) evaluating their predictive accuracy for human decision-making in an explore-exploit task; and 3) delivering these RNN-derived insights that traditional cognitive models overlook. Remarkably, our findings highlight the superiority of low-rank RNNs over alternatives like gated recurrent units in this task setting. More crucially, these RNNs reveal diverse strategies that individuals employ across different decision-making phases, advancing our understanding of intricate human decision-making processes. Our approach offers a powerful framework for discerning individual cognitive nuances.

## 1 Introduction

Just as the ebb and flow of neural networks in artificial intelligence, early researchers in psychology and neuroscience developed connectionist methods [1] to encapsulate the complexities of learning and decision-making in biological agents. In contrast, normative frameworks such as Bayesian inference [2, 3] and reinforcement learning [4, 5] currently gain the center stage for their interpretability and simplicity, but their handcrafted equations often limit the breadth of cognitive phenomena they could capture and inevitably introduce researchers' subjectivity into these cognitive models [6]. Aiming to overcome these issues, the recent renaissance of the connectionist approach directly fits artificial neural networks to observable behavior [6–9]. These networks offer high expressive power and can

---

[*]Co-first authors.
[†]Co-last authors.

NeurIPS 2023 AI for Science Workshop.

approximate a wide range of model structures by tuning the weights, thus facilitating the automatic discovery of behavioral strategies by analyzing these well-fitted models based on empirical data. To facilitate the interpretation of such networks, recent studies imposed different structural constraints on the networks to encourage the learning of a compact set of latent dynamical variables. However, each study only examined a particular architecture on a particular task, complicating direct comparisons of these architectures.

In this paper, we first provided a unified terminology for elucidating commonalities and differences among these architectures of low-dimensional recurrent neural networks (RNNs), including the low-rank RNN architecture [10–12], the tiny GRU architecture [6], and the disentanged RNN (disRNN) architecture [7]. Second, we compared them on a behavioral dataset of a two-armed bandit task designed for studying human explore-exploit behavior. We assessed their performance in capturing individual decision-making dynamics and found that low-rank RNNs outperform the others. Finally, from the view of dynamical systems, we applied logit analysis and symbolic regression to extract insights from the fitted low-rank RNNs and discovered a considerable diversity in cognitive algorithms that humans employ for value integration.

## 2 Comparing low-dimensional RNNs

Researchers have proposed to use tiny GRUs and disentangled RNNs to capture the computational mechanisms under sequential decision making [7]. In addition, the low-rank RNNs are introduced to understand the low-dimensional dynamics under perceptual decision making [10–12]. They embrace similar ideas by restricting the size of the bottleneck of the recurrent layer. Yet, It is less clear how these three architectures differ.

We here provide a unified terminology for comparing these networks. In each time step of these RNNs' dynamical equations, there are three separate pathways: the residual pathway ($h_t$) that maintains information (state variables) from previous time steps; the new pathway ($n_t$) that calculates the proposed update for the residual pathway based on the state $h_t$ and the input $I_t$; the gating pathway ($g_t$) that multiplicatively determines to what degree the new pathway should affect the residual pathway (e.g., a common updating rule is $h_{t+1} = (1 - g_t) \odot h_t + g_t \odot n_t$, where $\odot$ is the Hadamard product). These three architectures all limit the dimensionality of the residual pathway $h_t$, effectively constraining the amount of information that can flow across time steps.

### 2.1 Low-rank RNNs

We first transform the original formulas of low-rank RNNs into these three pathways.

The dynamics of a low-rank RNN with $N_n$ rate neurons are given by [12]

$$\tau \frac{dx_i}{dt} = -x_i + \sum_{j=1}^{N_n} J_{ij} \phi(x_j) + I_i^{ext}(t), \tag{1}$$

where $\tau$ is the time constant, $I_i^{ext}$ is the external input into neuron $x_i$, and $\phi(x)$ is a nonlinear activation function. Its discretized version is

$$x_{t+1}^{(i)} = (1 - \beta)x_t^{(i)} + \beta \sum_{j=1}^{N_n} J_{ij} \phi(x^{(j)}) + \beta I_i^{ext}, \tag{2}$$

where $\beta = dt/\tau$. Because the matrix $J$ is low-rank (rank $d$), following singular value decomposition, we have

$$J = \sum_{r=1}^{d} m^{(r)} n^{(r)T} = MN \tag{3}$$

where $m^{(r)}$ are the left singular vectors, corresponding to the columns of the matrix $M$, and $n^{(r)T}$ are the right singular vectors of $J$, corresponding to the rows of the matrix $N$.

Thus, Eq. 2 is transformed into (vector form)

$$x_{t+1} = (1 - \beta)x_t + \beta M \tilde{h}_t + \beta I^{ext}, \tag{4}$$

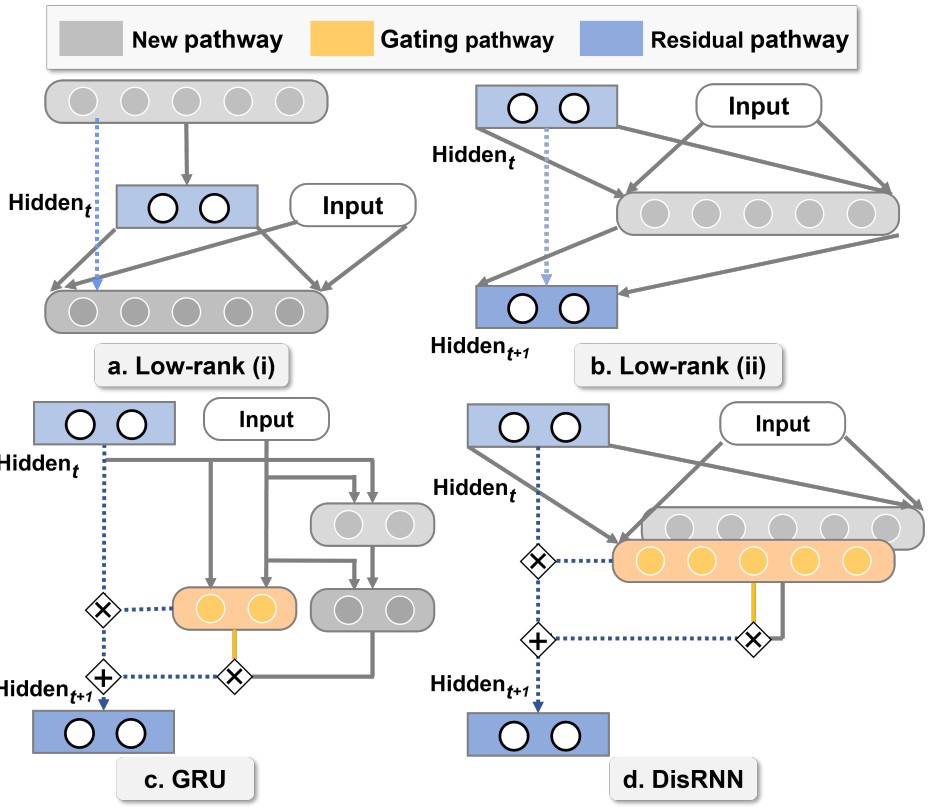

Figure 1: **Schematic of different low-dimensional RNNs**. **(a)** low-rank RNNs (i). **(b)**, low-rank RNNs (ii). **(c)** GRU. **(d)** disRNNs. **Grey**: new pathway; **yellow**: gating pathway; **blue**: residual pathway. Inputs consist of the action, reward, and other task information. These hidden states function as latent variables necessary for solving the task. Action probabilities are read out from the hidden states. Empty circles and colored circles are linear and nonlinear neurons, respectively.

where $\tilde{h}_t = N\phi(x_t)$ represents the bottleneck of update in the recurrent dynamics. $x_t$ undergoes the path $\phi(x_t) \xrightarrow[\text{compression}]{N} \tilde{h}_t \xrightarrow[\text{expansion}]{M} x_{t+1} \xRightarrow{\phi} \phi(x_{t+1})$ (Fig. 1a).

when $\beta = 1$, we have $x_{t+1} = M\tilde{h}_t + I^{ext}$. Equivalently, we can shift our focus on dynamics of the hidden state $h_t$ across time steps (equal to $\tilde{h}_t$ due to $\beta = 1$), shown in Fig. 1b. Here, the bottleneck $h_t$ undergo the path: $h_t \xrightarrow[\text{expansion}]{M} x_{t+1} \xRightarrow{\phi} \phi(x_{t+1}) \xrightarrow[\text{compression}]{N} h_{t+1}$, i.e., it is updated by

$$h_{t+1} = \tilde{h}_{t+1} = N\phi(Mh_t + I^{ext}). \tag{5}$$

We can add back a residual pathway

$$h_{t+1} = h_t + \tilde{h}_{t+1} = h_t + N\phi(Mh_t + I^{ext}), \tag{6}$$

or add back a residual pathway together with a constant ($= \beta$) gating pathway

$$h_{t+1} = (1 - \beta)h_t + \beta\tilde{h}_{t+1} = (1 - \beta)h_t + \beta N\phi(Mh_t + I^{ext}). \tag{7}$$

From the view of Eq. 5 (only the new pathway, or, equivalently considering a virtual residual pathway vanishes due to $\beta = 1$ gating pathway), Eq. 6 (both the residual and new pathways), Eq. 7 (all three pathways), the state $h_t$ of dimension $d$ can thus be treated as $d$ linear neurons (i.e., without nonlinear activation functions) in the network, while these expanded $x_t$ of dimension $N_n$ (previously considered rate neurons) are intermediate variables providing complex dynamics for transforming state $h_t$.

## 2.2 Tiny GRUs

The gated recurrent unit (GRU) [13] already contains three pathways (Fig. 1c). The hidden state $h_t$ is updated as follows:

$$
\begin{aligned}
r_t &= \sigma(W_{ir}u_t + b_{ir} + W_{hr}h_{t-1} + b_{hr}) \\
z_t &= \sigma(W_{iz}u_t + b_{iz} + W_{hz}h_{t-1} + b_{hz}) \\
n_t &= \tanh(W_{in}u_t + b_{in} + r_t \odot (W_{hn}h_{t-1} + b_{hn})) \\
h_t &= (1 - z_t) \odot n_t + z_t \odot h_{t-1}
\end{aligned}
\tag{8}
$$

where $\sigma$ is the sigmoid function, $\odot$ is the Hadamard product, $u_t$ and $h_t$ are the input and hidden state, and $r_t$, $z_t$, $n_t$ are the reset, update, new gates, respectively. Specifically, the hidden state $h_t$ (linear neurons) is the residual pathway, the $r_t$ and $n_t$ are two feedforward layers (with multiplicative interaction) in the new pathway, and the $z_t$ is the gating pathway. One limitation of the GRU architecture is that all three gates share the same dimensionality as the residual pathway $h_t$, limiting the complexity of temporal dynamics when there are only a few neurons available in $h_t$ (i.e., the dimensionality $d$ is low).

## 2.3 Disentangled RNNs

The original version of disentangled RNNs introduced a beta-VAE loss function and open/closed bottlenecks as a soft regularization on the dimensionality of the bottleneck [7]. Here, we focus on the disentangled RNNs with the same network architecture but without the beta-VAE loss. For structural comparison, we directly apply the hard regularization on the dimensionality of the bottleneck by limiting the number of neurons in it.

This architecture also consists of three pathways (Fig. 1d). The new pathway is updated by

$$
n_t = \text{MLP}_1(h_{t-1}, u_t),
\tag{9}
$$

while $u_t$ and $h_t$ are the input and hidden state (residual pathway), and MLP represents multi-layer perceptions (i.e. feedforward layers, showing one layer in Fig. 1d). The gating pathway is updated by

$$
g_t = \text{MLP}_2(h_{t-1}, u_t),
\tag{10}
$$

while $\text{MLP}_2$ can be either independent from or partially sharing weights with $\text{MLP}_1$. The hidden state $h_t$ (linear neurons) is updated by

$$
h_t = (1 - g_t) \odot n_t + g_t \odot h_{t-1}
\tag{11}
$$

This architecture is more flexible as the intermediate dimensionality in the MLP blocks could be higher than the dimensionality of $h_t$.

To summarize, neurons in residual pathways are linear in all three architectures and these three architectures can be compared via three pathways. Two versions of low-rank RNNs have three pathways but the gating pathway is simply a constant (one in Eq. 5 or non-one in Eq. 7). The other version of low-rank RNNs only has the residual and the new pathways. The tiny GRUs and the disRNNs have three pathways and their gating pathway are non-constant, depending on the inputs and the residual pathway. In addition, compared to tiny GRUs, the disRNNs have higher capacities in their new and gating pathways, providing more flexible modulation on the residual pathway.

## 3 Distilling individual decision-making dynamics

As we have analyzed the structural differences of these architectures, we next aimed to compare the performance of architectures trained on the same decision-making datasets. We specifically focused on a behavioral dataset collected from humans performing the Horizon task [14], designed for studying the explore-exploit behavior.

Our Horizon dataset comprises ~600,000 trials (100~1000 free trials per subject). In this task, participants engage in a sequence of games that each lasts for either 5 or 10 trials. In each game, participants choose between two slot machines that deliver probabilistic rewards sampled from a Gaussian distribution. The means of these distributions vary from game to game, making one option more rewarding on average. The standard deviations are 8 points. Without prior knowledge of the

Gaussian means, participants must explore both options to ascertain which one is the more rewarding choice. The first four trials in each game are "instructed trials", where players can only passively view an option and the associated reward. In some games, instructions are biased toward one option, creating a "high-information" scenario. In others, both options are equally presented, establishing a balanced information scenario. Following the instructed trials, participants make choices in either 1 (short-horizon) or 6 (long-horizon) "free trials". This influences the trade-off between exploration and exploitation, as the short horizon emphasizes exploitation due to limited future trials, and the long horizon encourages exploration as more future trials allow for capitalizing on explored information.

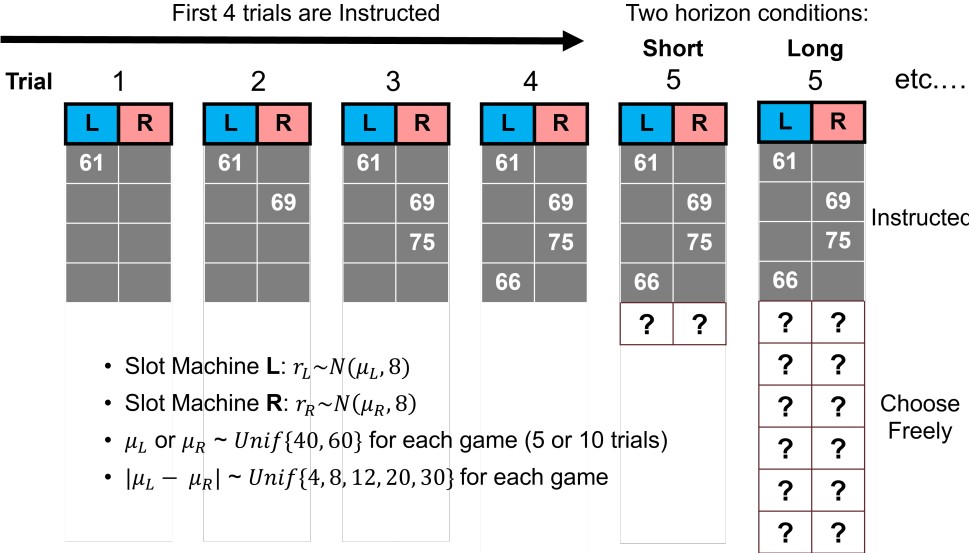

Figure 2: **The Horizon Task**. Each game begins with four instructed trials (grey square) in which participants are forced to pick the bandit with the green square. The number of instructed trials are identical for different horizon conditions. After the instructed trials, participants make free choices (white square) to the end of the game. In the short-horizon condition, the game ends after 1 free choice (4 instructed trials + 1 free trial = 5 trials). In the long-horizon condition, participants make 6 free choices to complete the game (4 instructed trials + 6 free trials = 10 trials).

To train low-dimensional networks for each subject to capture individual-specific strategies in such tasks, a common barrier is the limited number of trials per participant in psychological studies – often hundreds of trials per subject. In our Horizon dataset, for instance, each subject has approximately 100 to 1000 trials. Directly training a low-dimensional RNN model on one subject's choices will only result in worse model performance (results not shown).

Following the knowledge distillation framework (Fig. 3) proposed for addressing this issue [6], we first trained various larger teacher RNNs with different hyperparameters on the whole dataset of all subjects and selected the best models through nested cross-validation. We then trained a set of low-dimensional student RNNs for each individual to distill knowledge from the best teacher models, including tiny RNN, disRNN, and low-rank RNN architectures, each with distinct hyperparameters. We found that low-rank RNNs and the disRNNs performed better than the tiny GRUs in distilling individual decision-making dynamics (Fig. 4a). This suggests that the tiny GRUs, with gating and new pathways strongly limited by the number of available latent variables ($d$), cannot deal well with our task because of the task complexity such as long/short horizons and instructed/free trials. Low-rank RNNs achieved a performance better than (for $d = 1$) or similar to (for $d = 2$) the disRNNs, indicating the effectiveness of low-rank RNNs in low-dimensional scenarios. Upon selecting the optimal network for each subject based on each dimensionality, we observed that 1D student RNNs (light pink) outperformed the best logistic regression model (yellow) in [14], while 2D student RNNs (dark pink) approached the performance level of the teachers (grey). Removing dimensionality constraints in the selection process for each subject allowed the student RNNs (black) to match the performance of the teachers. These results suggest that low-rank RNN architecture is particularly

well-suited for capturing individual variability in our task setting. We will include more datasets in the future to examine the generalizability of its superior performance.

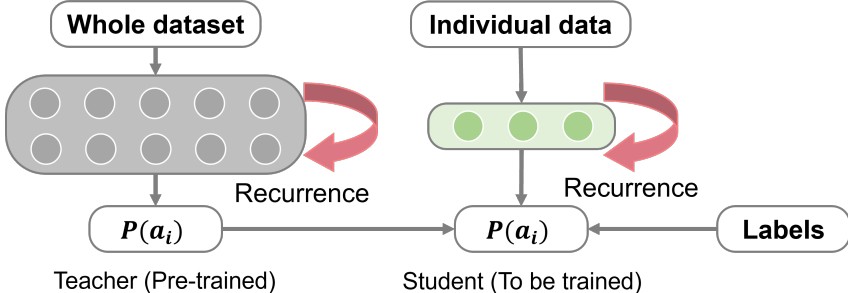

Figure 3: **Knowledge-distillation approach**. A larger teacher network is trained on the whole dataset of choices from all subjects. Subsequently, low-dimensional student networks are optimized to predict a weighted average of the teacher network's output action probabilities and ground truth choices (labels) on trials from a single subject.

## 4 Dimensionality of individual dynamics

The selected student RNNs effectively characterized the dimensionality of individual behavior, representing the degree of freedom a system needs to generate such behaviors. The dimensionality of an agent's behavior can be defined by the minimum number of latent variables needed for accurate future predictions [15, 6]. We determined the dimensionality of each subject's behavior based on the dimensionality of the student RNNs that achieved optimal performance in our nested cross-validation. We analyzed the distribution of behavioral dimensionality across all subjects, excluding 22 out of 643 subjects whose best-fitted models yielded a test accuracy below 60%. Our results show that the behavior of the majority of the subjects can be accurately captured using less than 5 dimensions (Fig. 4b).

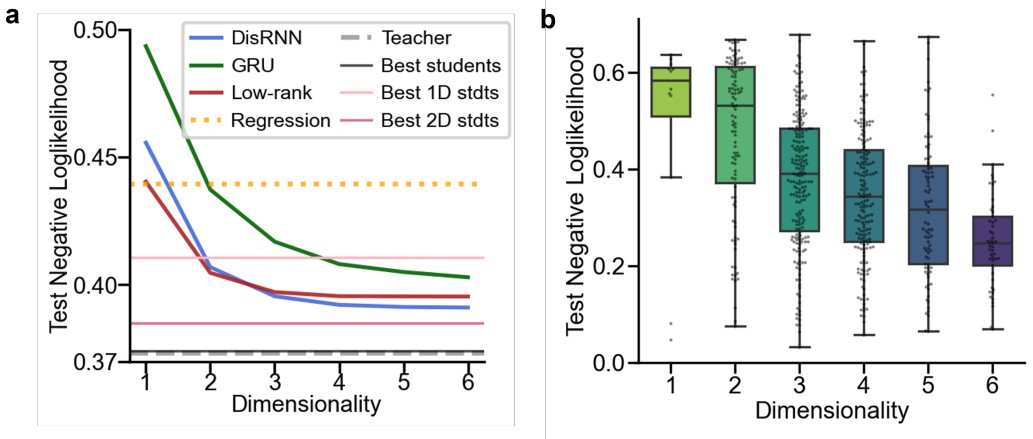

Figure 4: **Low-dimensional RNNs**. **(a)** The test performance (negative log-likelihood, the lower the better) of low-dimensional RNNs ("stdts" for students) and the best classical model ("Regression", studied in [14]). **(b)** The distribution of subjects' dimensionality of behaviors. Each point corresponds to a subject for whom that latent size achieved optimal performance.

## 5 Humans manifest diverse strategies to integrate values

As these low-dimensional RNN models have well captured the underlying behavioral patterns for each subject, we then aimed to identify what solutions these networks acquired – the cognitive strategies

used by each subject, by utilizing the dynamical systems interpretative framework developed by Ji-An et al. [6]. This methodology inspects the temporal changes of state variables (specifically, neuronal activations) in relation to inputs and the state variables themselves. A significant case of this approach is the logit analysis, mapping state variables and their changes at the output layer.

To define it mathematically, for a given trial $t$, the model's logit $L(t)$ is given by $L(t) = \log[\Pr_t(A_1)/\Pr_t(A_2)]$, where $\Pr_t(A)$ signifies the probability of selecting action $A$. A logit in the positive range suggests a preference towards $A_1$ over $A_2$, and vice versa. The change in logit, $\Delta L(t)$, is calculated as $L(t+1) - L(t)$, the difference in logits between successive trials. A positive (or negative) change in logit signifies an increased (or decreased) inclination for $A_1$ over $A_2$ due to the task input at trial $t$ (that is, the action followed by a reward).

To analyze these networks using the logit analysis, as the first step, we restricted our analysis to one-dimensional RNNs. There are two reasons for this choice: (i) a large proportion of variance in behavior can be captured by a one-dimensional RNN; (ii) in the one-dimensional case, there exists a one-to-one linear mapping between the logit (at the output layer) and the single hidden variable (at the recurrent layer), so the logit analysis fully captured the dynamics of the hidden variable.

For each subject in each phase (free/instructed), we treated the logit change $\Delta L(t)$ (z-axis) as a function of the current logit $L(t)$ (x-axis) and reward (y-axis), for all trials selecting $A_1$ (see Fig. 5). This function effectively summarizes how the logit (i.e., the preference for $A_1$) will change in the next trial due to the selected action $A_1$ and the associated reward. We utilize symbolic regression [16, 17] to learn this function for better visualization purposes. The mean squared loss of symbolic regression for all subjects was less than 0.01, indicating a reasonably good approximation.

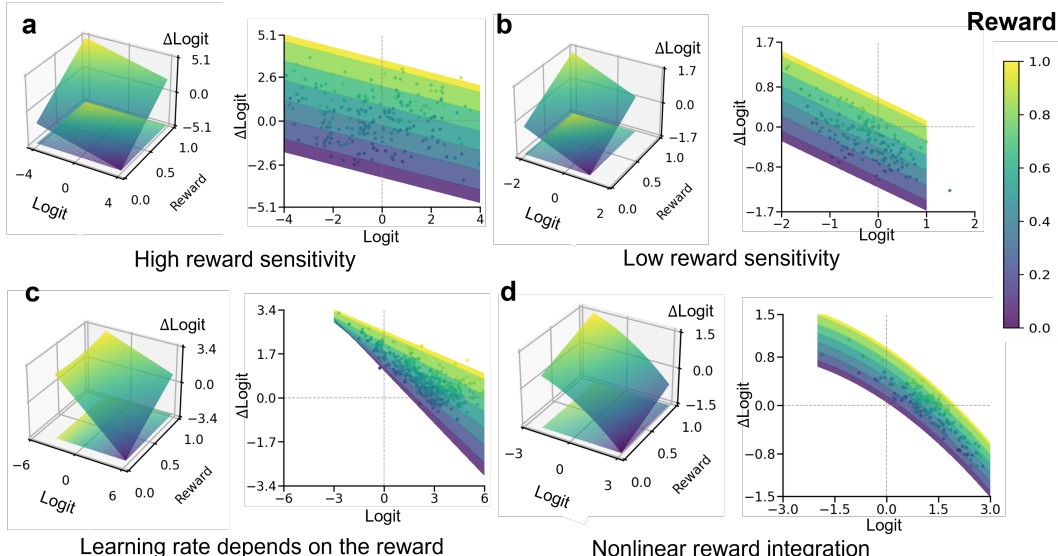

Figure 5: **Individual behavioral dynamics**. On the **left** side of each panel, the logit change ($\Delta$logits, z-axis) as a function of the current logit (x-axis) and trial-associated reward (y-axis). On the **right** side of each panel, the x-axis represents the logits and the y-axis represents the logit changes, obtained by projecting the 3D figure onto the x-z plane. The colored reward stripes demarcate varying levels of received rewards. Each point represents a trial, colored by the associated reward.

We then projected the 3D surface to colored reward stripes in the x-z plane and plotted the logit changes versus logits for each trial, with the trials colored by the associated rewards. We note that these visualizations offer rich characterizations of the dynamical evolution of the system (for a comprehensive introduction see Ji-An et al. [6]), while we only point out a few patterns revealed by these figures.

(i) The negative slope $s$ ($s < 0$) of the reward stripe is directly related to the concept of learning rate $\alpha$ in classical reinforcement learning models ($\alpha = -s$). The first subject (Fig. 5a) has a larger learning rate ($\alpha = 0.42$) than the second (Fig. 5b, $\alpha = 0.34$).

(ii) The intercept of the reward stripe onto the $x = 0$ line directly reflects the magnitude of the subjectively perceived reward, similar to the utility functions of the reward (refer to the asymptotic preference in [6]). The larger range of the reward stripes projecting onto the $x = 0$ in Fig. 5a (compared to Fig. 5b) indicates a larger reward sensitivity.

(iii) For some subjects (e.g., Fig. 5c) we found that the learning rate (negative slope) of the reward stripes indeed depends on the reward magnitude, with a larger learning rate corresponding to a smaller reward. This tendency is not revealed by any cognitive models studied for this task, and we discovered it by simply visualizing these logit patterns.

(iv) Moreover, we observed non-linear evidence-integration in some subjects (e.g., Fig. 5d).

Overall, by analyzing these one-dimensional RNNs using the logit analysis, we discovered diverse and heterogeneous individual differences that are difficult to discern for classical cognitive models. Identifying the best architectures and analyzing these well-trained low-dimensional models in detail, will substantially advance our understanding of these complex patterns underlying biological decision-making behavior.

## 6 Conclusion and future work

The intricacies of decision-making in biological agents, especially humans, have long been the subject of study in psychology and neuroscience. Recent studies in RNNs provide a promising avenue for understanding these processes with reduced human bias and greater predictive accuracy. Our study dives deep into the architecture of low-dimensional RNNs, offering a standardized perspective that aids in discerning their differences and potentialities. Remarkably, the low-rank RNNs emerge as superior models for capturing individual nuances in the context of the explore-exploit task. Our approach further elucidates the latent dynamical variables governing decision-making in this task, highlighting the diverse strategies individuals employ.

Going forward, several avenues remain unexplored. Future research will investigate the generalizability of the performance of these low-dimensional RNNs in diverse population cohorts across different types of decision-making tasks that might involve other cognitive functions like attention and memory. Furthermore, we will keep developing visualization techniques for understanding higher-dimensional RNN models. We are optimistic about the marriage of computational cognitive science with neural network interpretability in reshaping our understanding of human decision-making at both individual and collective levels.

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
