# OpenReview forum: "Distilling human decision-making dynamics: a comparative analysis of low-dimensional architectures"
_NeurIPS.cc/2023/Workshop/AI4Science — NeurIPS2023-AI4Science Poster_

### Official Review · Reviewer_bYG7 · 2023-10-20
**very interesting paper**

**Rating:** 8
**Confidence:** 3

**Review:**

The paper aims to delve deeper into the relevance and effectiveness of low-dimensional RNN architectures in comprehending biological decision-making behaviors. The focus is mainly on the differences in their practical effectiveness and how they compare with traditional cognitive models.

Strengths:

By standardizing terminology and providing a broad comparison of low-dimensional RNN architectures, the authors make an important contribution to the domain, aiding researchers and practitioners alike.

The paper doesn’t merely compare models but goes a step further in deducing insights that conventional cognitive models might have missed. The identification of diverse strategies individuals use across decision-making phases is an invaluable addition to the literature.

---

### Official Review · Reviewer_vBpH · 2023-10-24
**Review of Submission173**

**Rating:** 4
**Confidence:** 3

**Review:**

__Summary__: The work discusses post-hoc interpretation methods for analyzing recurrent models on human decision making. The authors provide a unifying framework for several major classes of recurrent models to compare them from an architectural standpoint. The authors then apply low-dimensional versions of these RNN models to interpret decisions and learning rates in a multi-arm bandit task.

__Strengths__:
- For readers unfamiliar with RNN architectures, the unifying characterization seems helpful and succinct.
- The logit vs. delta-logit comparison of decisions is interpretable and mathematically oriented, and provides an intuitive way to understand model learning rates and biases in the bandit task.
- It may be interesting that the performance of student models can be improved by repeated runs of a teacher model on the same data, although this seems to resemble bootstrapping.

__Weaknesses__:
- It is unclear if the experimental setup can really be used to analyze human decisions. The authors make a fundamental assumption that a singular large RNN can capture the dynamics of all individual human decision processes, and that experiment-subject-specific decision processes are identified by passing subject-specific data to this RNN. This seems like an unrealistic assumption, given the authors aim to interpret individual decision processes as students of this teacher model, and presents a major confounding variable for the post-hoc analysis.
- As mentioned above, it is unclear if the teacher-student modeling regime actually improves performance or if this could be achieved using bootstrapping. The teacher model in general seems to limit interpretation of the student models.
- The bandit setting is simple and there seems to be only a small amount of information to distill about human decisions from this highly controlled task on mean estimation. Interpreting decision-making would be most interesting in complex environments like medical decision-making, where decision models could lead to improved hospital operations or patient outcomes. Pace et al. (2022) address this in their POETREE work, among other related works.
- The logit analysis only appears to work in a univariate setting, and as such has limited utility. If this could be extended to a multi-action setting with many hidden states, this post-hoc interpretability approach would be much more impactful.
- The comparison of RNN architectures is mostly unrelated to the discussion of decision-making dynamics, and only the architecture with the best performance is evaluated. It would be more interesting and fair to compare these architectures on a wide variety of tasks to interpret how human decisions may align better with each of these models in different situations.

__Nits:__
- Biological decision making is a misleading term. It seems the authors mean to say human decision making.
- Learning rates are mentioned, but not explored. It would be interesting to characterize human decisions relative to known theoretical criteria (e.g. optima, smoothness).


__Recommendation__:
The authors identify an important problem and present a clear need for interpretability methods for black-box models of human decision making. However, there seem to be fundamental flaws that limit the interpretation of the results, as well as the impact of the work, and I cannot recommend acceptance of the work in its current state.